# Integrative Analysis of Metabolome and Proteome in the Cerebrospinal Fluid of Patients with Multiple System Atrophy

**DOI:** 10.3390/cells14040265

**Published:** 2025-02-12

**Authors:** Nimisha Pradeep George, Minjun Kwon, Yong Eun Jang, Seok Gi Kim, Ji Su Hwang, Sang Seop Lee, Gwang Lee

**Affiliations:** 1Department of Molecular Science and Technology, Ajou University, Suwon 16499, Republic of Korea; nimishapgeorge@ajou.ac.kr (N.P.G.); kmj936@ajou.ac.kr (M.K.); jye120@ajou.ac.kr (Y.E.J.); rlatjrrl9977@ajou.ac.kr (S.G.K.); js3004@ajou.ac.kr (J.S.H.); 2Department of Physiology, Ajou University School of Medicine, Suwon 16499, Republic of Korea; 3Department of Pharmacology, Inje University College of Medicine, Busan 50834, Republic of Korea; leess@inje.ac.kr

**Keywords:** cerebrospinal fluid, in silico prediction, integrated omics, metabolomics, multiple system atrophy, proteomics

## Abstract

Multiple system atrophy (MSA) is a progressive neurodegenerative synucleinopathy. Differentiating MSA from other synucleinopathies, especially in the early stages, is challenging because of its overlapping symptoms with other forms of Parkinsonism. Thus, there is a pressing need to clarify the underlying biological mechanisms and identify specific biomarkers for MSA. The metabolic profile of cerebrospinal fluid (CSF) is known to be altered in MSA. To further investigate the biological mechanisms behind the metabolic changes, we created a network of altered CSF metabolites in patients with MSA and analysed these changes using bioinformatic software. Acknowledging the limitations of metabolomics, we incorporated proteomic data to improve the overall comprehensiveness of the study. Our in silico predictions showed elevated ROS, cytoplasmic inclusions, white matter demyelination, ataxia, and neurodegeneration, with ATP concentration, neurotransmitter release, and oligodendrocyte count predicted to be suppressed in MSA CSF samples. Machine learning and dimension reduction are important multi-omics approaches as they handle large amounts of data, identify patterns, and make predictions while reducing variance without information loss and generating easily visualised plots that help identify clusters, patterns, or outliers. Thus, integrated multiomics and machine learning approaches are essential for elucidating neurodegenerative mechanisms and identifying potential diagnostic biomarkers of MSA.

## 1. Introduction

Multiple system atrophy (MSA) is a rare, adult-onset, progressive neurodegenerative disorder belonging to the synucleinopathy family. It is characterised by the abnormal accumulation of α-synuclein in oligodendrocytes, which forms distinctive glial cytoplasmic inclusions (GCIs) known as Papp-Lantos bodies [1,2,3,4]. Clinically, MSA manifests with a range of symptoms, including autonomic dysfunction, Parkinsonism, and cerebellar ataxia, making its differential diagnosis challenging owing to its overlap with other neurodegenerative diseases [5,6,7,8]. As MSA progresses more aggressively than other synucleinopathies and has a limited response to symptomatic treatment, early and accurate diagnosis is critical for treatment and symptom management [9]. Unfortunately, no current disease-modifying therapies or accurate biomarkers are available for MSA, and symptomatic management remains limited.

As a major strategy for analysing new biomarkers and mechanisms, metabolomic approaches have facilitated the discovery of novel metabolic biomarkers for diagnosing MSA’s complex symptoms, especially through CSF [10], which is the most relevant biological fluid because of its proximity to the brain and its reflection of the pathophysiological state of the disease. For example, metabolic profiling of free fatty acids and polyamines in the CSF has proven useful in distinguishing MSA from Parkinson’s disease (PD) in our previous studies [11,12], aiding in understanding the disease mechanisms of MSA and uncovering clinically relevant biomarkers. Elevated levels of the free fatty acid eicosapentaenoic acid (EPA) were observed in the CSF of both MSA and PD patients compared to healthy controls. However, no significant difference was found between MSA and PD, emphasizing the need for further metabolomic studies to distinguish MSA from other synucleinopathies, which is critical for implementing appropriate treatment strategies at the onset of each disease.

Because metabolomes can effectively showcase phenotypic traits, while proteomes provide a dynamic representation of cellular states and direct functional insight into CSF [13], there is a critical need to analyse changes in the metabolome and proteome across different body fluids, especially the CSF, which flows through the ventricular system of the brain. CSF is considered significantly valuable for studying MSA because of its direct connection to the central nervous system (CNS), making it uniquely suited to reflect the brain pathophysiology. CSF is considered a promising fluid for biomarker discovery in neurodegeneration, particularly because there are limited studies on biomarkers specific to MSA. Consequently, additional studies that analyse alterations in the CSF metabolome and proteome through both experimental and bioinformatic approaches to gain insights into MSA-related pathophysiological mechanisms are imperative.

To achieve an in-depth understanding of complex biological processes, it is essential to integrate various omics data to emphasise the interactions between biomolecules and their functions through comprehensive analysis [14]. The integration of two or more omics layers is a prominent approach in elucidating the mechanism of neurovegetative diseases and advancing biomarker discovery precision. For instance, Hwang et al. recently integrated miRNAome and proteome in their analysis of CSF in patients with PD [15]. However, these inherently high dimensional data escalate the convolutions of data interpretation. Therefore, to tackle such challenges, dimensionality reduction and clustering are employed to eliminate redundant or irrelevant information while simplifying the data for clearer analysis and interpretation [16]. These multiomics approaches enable the capture of more complex biological interactions [17,18,19], allowing the identification of novel molecular signatures that reflect disease mechanisms. In addition, multiomics data, together with big data and computational approaches, are helpful for handling vast amounts of large-scale omics data [20,21,22,23]. In this review, we aim to (i) characterise the features of MSA, (ii) biochemically analyse CSF for the diagnosis of MSA, (iii) apply metabolomic approaches in CSF with MSA, (iv) analyse proteomes in CSF with MSA, (v) integrate omics approaches to enhance diagnostic accuracy, and (vi) discuss future perspectives using machine learning.

## 2. Characteristic Features of MSA

MSA, formerly called Shy–Drager syndrome, is a rare and rapidly progressive but severe neurodegenerative disorder with a prevalence of 3–5 per 100,000 [24,25] and an incidence of 0.6–3/100,000 people per year [26,27], which increases with age, and the average age of onset is in the sixth decade [28]. This affects both males and females without any distinction. The median survival of patients with MSA is 6–10 years from symptom onset [29], underscoring the importance of early and precise diagnosis. MSA is a fatal neurodegenerative disease compared to other types of Parkinsonism. Differentiating MSA from other forms of atypical parkinsonism and progressive neurodegenerative disorders remains challenging, especially during the early stages of the disease. Many of the symptoms and pathophysiologies, such as motor dysfunction, autonomic dysfunction, and cognitive decline, overlap with those of other neurodegenerative disorders, complicating the accurate diagnosis of MSA.

MSA is pathologically characterised by the progressive loss of neurons and gliosis in specific brain regions, including the basal ganglia, cerebellum, pons, and spinal cord [30]. The Movement Disorder Society classifies MSA into four types based on diagnostic criteria: neuropathologically established MSA, clinically established MSA, clinically probable MSA, and possible prodromal MSA. The widespread and abundant spread of α-synuclein-positive GCIs, along with striatonigral or olivopontocerebellar neurodegeneration, implies neuropathologically established MSA. Autonomic dysfunction accompanied by Parkinsonism and/or cerebellar syndrome falls under probable and possible MSA [31,32].

MSA can be divided into two subtypes: multiple system atrophy—Parkinsonian type (MSA-P) and multiple system atrophy—cerebellar type (MSA-C). Primary symptoms of MSA-P resemble Parkinson’s disease such as tremors, rigidity and bradykinesia while MSA-C primarily affects the cerebellum, impairing balance and causing coordination issues in patients [33]. While most reports present MSA as a general disease and do not differentiate between the two subtypes, Abdo et al. attempted to differentiate between the two based on neurochemical profile and reported no distinction between MSA-P and MSA-C. This highlights the importance of additional studies required for a deeper understanding of MSA and to distinguish between MSA-P and MSA-C [34].

Several studies on MSA pathology have highlighted numerous contributing factors, such as genetic mutations of α-synuclein [35,36], protein misfolding [37], oxidative stress [38], myelin dysfunction [39], and mitochondrial dysfunction [40,41,42]. The hallmark of MSA is oligodendroglial cytoplasmic inclusions rich in aggregated α-synuclein [43], although no definitive biomarkers for MSA have been identified. Oligodendrocytes are specialised glial cells in the CNS responsible for producing and maintaining myelin, the protective sheath around axons that ensures efficient nerve signal transmission. When oligodendrocytes are damaged or lost, demyelination occurs, leading to disrupted neural communication and subsequent neurological deficits. In addition to the notable GCIs in the MSA brain, α-synuclein aggregation has also been reported in oligodendrocyte nuclei (GNI), neuronal cytoplasm (NCI), and neuronal nuclei (NNI). Notably, the distribution of GCIs and GNIs was more prominent than that of NCIs and NNIs across the brain in connection with neurodegeneration, indicating the pivotal role of oligodendrocytes in neurotoxicity. Hence, oligodendrogliopathy is a primary pathological cause of MSA [25,31,44]. As our understanding of the molecular mechanisms of MSA advances, there is a growing demand for the development of new diagnostic markers.

## 3. Biochemical Analysis of CSF for the Diagnosis of MSA

Since CSF has been considered a “neurochemical window” in CNS, a lot of studies have been carried out to investigate altered biomolecules for the early diagnosis of MSA, such as proteins including α-synuclein, beta-amyloid 42 (Aβ42), total tau (t-tau), neurofilament light chain (NfL), tumour necrosis factor (TNF) [45,46,47], microRNA [48,49], and metabolites [50,51,52]. Approximately 20% of the proteins in the CSF originate specifically from brain cells, including neurons, astrocytes, and glial cells, whereas approximately 80% of the proteins in the CSF originate from the filtration of the peripheral blood [53,54]. This brain-specific component makes CSF a more precise indicator of CNS changes than other fluids, such as blood, saliva, and urine. To analyse the activity of the CNS pathway in patients with MSA, the levels of neurotransmitters, such as 3-methoxy-4-hydroxy-phenylglycol (MHPG), 5-hydroxyindoleacetic acid (5-HIAA), and homovanillic acid (HVA), in the CSF were reduced in patients with MSA compared to healthy controls [52]. In particular, the levels of MHPG and 5-HIAA in the CSF are significantly lower (by 49–70%) in MSA than in PD [10]. Although the CSF is considered an optimal fluid for biomarker discovery in neurodegeneration, studies specific to MSA remain sparse and sometimes inconsistent. However, the acquisition of additional data on CSF molecules related to MSA combined with the application of advanced bioinformatics software and machine learning is anticipated to enable the identification of more precise cellular biological mechanisms and robust biomarkers of MSA in the future.

## 4. Metabolomics Approaches in CSF with MSA

Metabolomics, an indispensable biochemical framework, measures changes in primary and secondary metabolites in the CSF, thereby providing deeper insights into metabolic disruptions that accompany MSA. Primary metabolites such as amino acids and lipids are critical for energy production, neurotransmitter synthesis and structural integrity for neurons. They are produced through fundamental metabolic pathways such as the TCA cycle. Secondary metabolites such as neuroactive peptides serve specialized roles in neuroprotection, signalling, and adaptation to stress and are derived from primary metabolites or complex pathways. In our previous reports, the level of the primary metabolite EPA, which has an anti-inflammatory function, was increased in the CSF of patients with MSA [11]. Secondary metabolites, polyamines (PAs) in the CSF of the MSA group, and concentrations of *N*^1^-acetylcadaverine, cadaverine, *N*^8^-acetylspermidine, and total PAs were significantly increased, while concentrations of *N*^1^-acetylputrescine and *N*^1^-acetylspermidine were significantly reduced compared to the normal group [12].

Metabolic changes in the CSF of patients with MSA were analysed using ingenuity pathway analysis (IPA) (http://www.ingenuity.com, accessed on 17 December 2024) [55]. IPA is a web-based bioinformatics software that allows researchers to visualise complex omics data, perform data analysis, and understand data in a biological context. For the data acquisition process, we conducted a systematic literature search using the terms “biomarkers”, “metabolite” or “metabolome”, “MSA” or “multiple system atrophy”, and “CSF” or “cerebrospinal fluid” in PubMed and Google Scholar. These search terms were applied in various combinations to ensure comprehensive coverage of the available literature. Through this process, we identified 16 studies reporting metabolite biomarkers in MSA patients compared to healthy controls using CSF samples. To ensure a rigorous and unbiased selection of studies, we adhered to a set of inclusion and exclusion criteria. The inclusion criteria were studies reporting metabolites with statistically significant fold changes between MSA patients and healthy controls, studies that provided explicit fold-change data or other quantifiable measures for comparison and comparisons restricted to MSA patients and normal healthy controls. Exclusion criteria for this study excluded metabolites without reported fold-change values or statistical significance, studies including comparisons with disease groups other than MSA or controls and reports where metabolite changes were not statistically validated. During data acquisition, we identified 26 primary and secondary metabolites linked to MSA in the CSF. Of these, only 19 metabolites showed a statistically significant difference in their metabolic profiles compared to healthy controls; therefore, only these molecules were uploaded to the IPA program for analysis and network construction (Table 1 and Appendix A, Figure 1A). Details of the remaining seven molecules are listed in Appendix A. From the broad range of functions and diseases available for prediction in the IPA program, five functions and three diseases representing key MSA hallmarks were selected. Since the predictions are based solely on a meticulously curated dataset, incorporating fold changes and statistical significance of molecular expression alterations in MSA patients compared to healthy controls, this approach minimizes any potential bias that might arise from subjective selection of functions and diseases. Among these 19 metabolites, 6 showed elevated expression levels, whereas the remaining 13 were downregulated. Based on these measurements, two biological functions, ATP concentration and neurotransmitter release, were predicted to be inhibited, while neurodegeneration was predicted to be activated (Figure 1B). Lactic acid, coenzyme Q10 (CoQ10), 3,4-dihydroxyphenylalanine (L-dopa), dopamine, norepinephrine, and neuropeptide Y have emerged as key players in predicting functions and diseases in the network.

In our network, we observed that an increase in lactic acid concentration was directly associated with a decrease in the ATP concentration. Elevated levels of lactic acid have been reported to impair ATP synthesis by creating an oxidising environment, which in turn leads to the degeneration of neurons [61,62,63]. This aligns with our in silico prediction that shows increased lactic acid levels directly impact ATP levels (Appendix A). CoQ10 is a powerful antioxidant that regulates cellular function and is responsible for the maintenance of the mitochondrial ETC. Consequently, CoQ10 deficiency leads to impaired ATP production, increased oxidative stress, and neurodegeneration [64]. Several reports have previously established substantially reduced levels of CoQ10 in the plasma, CSF, and cerebellum of patients with MSA compared to healthy controls [58,65,66,67]. This aligns with our in silico prediction that downregulation of CoQ10 can lead to a diminished concentration of ATP and further trigger neurodegeneration. Interestingly neurotransmitters, such as dopamine, dopamine precursor L-dopa, norepinephrine-e, and neuropeptide Y, displayed lower levels and were linked to the inhibition of neurotransmitter release and amplified neurodegeneration. Dopamine, a catecholamine neurotransmitter, is involved in motor functions, cognition, memory, and other brain functions. As a result, dysfunction or suppressed release of dopamine can lead to mitochondrial dysfunction, impaired respiratory chain activity, neuronal damage, and neuroinflammation [68,69].

Collectively, the modulated expression of these metabolites indicates biological mechanisms leading to neurodegeneration. Regrettably, the concentrations of ROS, oligodendrocytes, cytoplasmic inclusions, white matter demyelination, and ataxia were not associated with any of the metabolites in the network. To overcome the limited number of metabolites and obtain more comprehensive information on molecules that change in CSF in MSA patients, we extended our analysis to include proteomic profiling of altered proteomes in the CSF of patients with MSA.

## 5. Analysis of Proteomes in CSF with MSA

Proteomics is a comprehensive, large-scale study of proteins [70,71] that began with the introduction of 2-dimensional electrophoresis (2-DE) and protein mapping in bacteria in 1974 [72]. Although metabolomics provides quantitative information on metabolic alterations and accurately represents the phenotype, proteomic analysis offers complementary information on molecular processes and provides more molecules than metabolomics. Proteomic approaches can provide comprehensive information on protein-level changes that drive the progression of neurodegenerative diseases [73]. Owing to the development of analytical instruments, proteomics has been used to analyse altered proteins in the brain related to neurodegenerative diseases, such as Alzheimer’s disease, PD, and amyotrophic lateral sclerosis [73,74,75,76]. Compared to methods such as 2-DE, enzyme-linked immunosorbent assay (ELISA), and antibody arrays, liquid chromatography–tandem mass spectrometry (LC-MS/MS) is a powerful and versatile analytical technique widely used in proteomics, capable of detecting a wide range of biomolecules, including peptides, proteins, lipids, metabolites, and post-translational modifications, as well as having high sensitivity and specificity for detecting and quantifying analytes, even at very low concentrations (e.g., femtomolar range) [77,78]. Thus, LC-MS/MS is a good approach for analysing the CSF in MSA [79].

For the data acquisition processes, the search terms used in different combinations were “biomarkers”, “protein”, “proteomics”, “MSA or multiple system atrophy”, and “CSF or cerebrospinal fluid” in PubMed and Google Scholar. These keywords were used in different sequences to obtain a thorough exposure of the available literature. Through this process, we identified 31 studies reporting protein biomarkers in MSA patients compared to healthy controls using CSF samples. To ensure a thorough and impartial selection of studies, we followed predefined inclusion and exclusion criteria. The inclusion criteria were studies reporting metabolites with statistically significant fold changes between MSA patients and healthy controls, studies that provided explicit fold-change data or other quantifiable measures for comparison, studies employing whole CSF samples for protein profiling, and comparisons restricted to MSA patients and normal healthy controls. Exclusion criteria for this study excluded metabolites without reported fold-change values or statistical significance, studies including comparisons with disease groups other than MSA or controls, studies using fractionated CSF samples to study protein concentration in individual fractions, and reports where metabolite changes were not statistically validated. We identified 110 proteins as potential biomarkers in CSF samples of patients with MSA. Among them, only 47 proteins showed statistically significant changes in their expression levels in patients with MSA compared with healthy controls; therefore, only these proteins were included in the dataset for the IPA (Table 2 and Appendix A). The details of the remaining proteins and their expression levels are presented in Appendix A. Of the 47 proteins used for the IPA, 19 were upregulated, and 28 were downregulated (Figure 2A). Three of the eight biological functions (concentration of ATP, release of neurotransmitters, and number of oligodendrocytes) were predicted to be inhibited, whereas the concentration of ROS, number of cytoplasmic inclusions, demyelination of white matter, ataxia, and neurodegeneration were predicted to be inhibited (Figure 2B). Among these, complement C3 (C3), microtubule-associated protein tau (MAPT), nerve growth factor (NGF), α-synuclein (SNCA), macrophage colony-stimulating factor (CSF1), glial fibrillary acidic protein (GFAP), and Parkinson’s disease protein 7 (PARK7) were linked to two or more functions.

As MSA cannot be identified by one protein or symptom alone, we focused on proteins with a direct link to functions that highlighted the characteristic features of MSA, mainly oligodendrocyte number, amount of cytoplasmic inclusions, and white matter demyelination. CSF1 expression was decreased and correlated with a reduced number of oligodendrocytes in our network. Liu et al. performed both ex vivo and in vivo experiments on cerebellar explants from mice and in purified primary murine culture, respectively, to study the effect of inhibition of CSF1 on oligodendrocyte precursor cells (OPCs). They quantified the number of OPCs using immunostaining methods and established that inhibiting CSF1 directly impaired OPC viability in vitro and caused a reduction in OPC numbers ex vivo and in vivo [95]. As CSF1 is predominantly expressed in the white matter, this aligns with our in silico prediction that decreased CSF1 levels lead to a decrease in the number of oligodendrocytes [96]. Interestingly, upregulated GFAP levels directly increase the formation of cytoplasmic inclusions and neurodegeneration. GFAP is emerging as a potential indicator of reactive astrogliosis, which is associated with the clinical disease severity in MSA. Astrocytes display high GFAP expression and hypertrophic astrocytic processes in close proximity to the GCI in MSA tissue. This activation was observed to decrease as astrocytes moved further away from the GCI [21]. Additionally, demyelination of the white matter was predicted to be directly elicited by increased IL10 levels in the samples. Smith et al. [97] and Dace et al. [98] have confirmed demyelination due to IL10 overexpression in different neuropathy models. Taken together, a decrease in the number of oligodendrocytes, along with increased cytoplasmic inclusions, demyelination, and ataxia, leads to atrophy of multiple systems.

The proteomic network was able to link and predict all biological functions, that is, concentration of ATP, concentration of ROS, quantity of oligodendrocytes, quantity of cytoplasmic inclusions, demyelination of white matter, ataxia, and neurodegeneration. This evaluation of the protein network revealed connections between CSF proteins and their potential roles in MSA, offering a more comprehensive understanding of MSA pathology.

## 6. Integrated Omics to Enhance Diagnostic Accuracy

By integrating different omics layers, researchers can obtain a more comprehensive understanding of complex biological systems, identify biomarkers, understand disease mechanisms and pathways, and overcome the limitations of single omics studies by providing a multifaceted view of cellular processes.

Combining metabolomics and proteomics enables a better understanding of the molecular processes occurring in the CSF of patients with MSA using IPA. To obtain an integrated metabolomic–proteomic network, the metabolites and proteins were combined and entered into the IPA program for analysis (Figure 3A, Appendix A). Since the IPA program makes predictions based on a curated dataset, it helps minimize potential bias. The enrichment of neuron-related pathways underscores the essential role of these biomarkers in the pathophysiology of MSA. Based on the integrated metabolome–proteome network, we observed that an increase in lactic acid, which contributes to a reduced concentration of ATP, is associated with increased levels of programmed cell death 1 ligand 1 (CD274), interleukin-8 (CXCL8) and alpha-1-antichymotrypsin (SERPINA3) and decreased levels of chromogranin-A (CHGA). Although not in a true neurological context, the abundance of CD274 protein as a result of decreased ATP levels has been established [99]. Similarly, CXCL8 [100] and SERPINA3 [101] have been implicated in the suppression of ATP levels in inflammatory pathways. Lower dopamine expression was responsible for the suppressed release of neurotransmitters and activated neurodegeneration and was found to be linked with monocyte chemoattractant protein-1 (CCL2), macrophage-derived chemokine (CCL22), IL-8 (CXCL8), transforming growth factor-α (TGFA), and vascular endothelial growth factor A (VEGFA). Integrated omics was able to predict all eight functions consistent with single omics analysis and, therefore, overcome the limitations of the single metabolome network (Figure 3B).

The top 10 neurological diseases and biological functions according to the activation z-score predicted to be inhibited (Figure 3B, Appendix A) and activated (Figure 3C, Appendix A) are shown. The top diseases and biological functions predicted to be suppressed were the development of neurons, the density of neurons, and the release of transmitters. The maximum number of molecules consisting of different proteins and metabolites are involved in diverse pathways, such as oxidative stress, mitochondrial dysfunction, protein aggregation, excitotoxicity, and neuroinflammation, leading to the inhibition of neuronal cell death. Familial and progressive neurological disorders are among the top ten activated diseases and their biological functions. Neuronal firing ranked second in terms of activated diseases and biological functions. Several studies have implicated abnormal neuronal firing in neurotoxicity in both mouse and rat models [102,103,104].

Metabolomic and proteomic analyses of CSF in MSA have provided valuable insights; however, the limited sample size remains a significant limitation. Incorporating additional omics approaches, such as transcriptomics and phosphoproteomics, is necessary to expand future multiomics research. This integrated approach enables researchers to identify patterns across different molecular levels as like our previous reports [20,105,106,107,108] and offers deeper insights into the disease. Thus, it is very important to incorporate “dimensionality reduction” and “machine-learning clustering algorithms”.

## 7. Future Directions Using Machine Learning

Multiomics data are high dimensional, complex, and prone to noise, posing computational challenges [14,16]. An effective analysis requires dimensionality reduction techniques to manage data complexity and clustering-based approaches to identify meaningful patterns [109,110] (Figure 4).

Dimensionality reduction includes methods such as Principal Component Analysis (PCA), t-distributed Stochastic Neighbor Embedding (t-SNE), and Uniform Manifold Approximation and Projection (UMAP), which facilitate the simplification of complex datasets while retaining essential information. These methods can significantly reduce noise and extract meaningful biological insights from complex multiomics data. Clustering algorithms are broadly categorised based on their underlying assumptions, including partitioning, density, and hierarchical approaches [111]. For instance, k-means clustering is a partition-based method that aims to minimise the sum of squared distances between data points and their assigned cluster centroids. It is particularly effective for datasets with spherical clusters and computationally efficient for large datasets [112]. Density-based methods such as DBSCAN identify clusters as regions of high density separated by low-density areas, making them suitable for data with noise and irregular cluster shapes [113]. HDBSCAN extends DBSCAN by incorporating hierarchical clustering, allowing the discovery of clusters with varying densities without requiring the number of clusters to be specified beforehand [114]. OPTICS further improves DBSCAN by ordering data points based on density, enabling the detection of clusters with varying densities without a fixed distance threshold [115]. A silhouette score was used to quantitatively assess the quality and stability of the clusters generated by DBSCAN, HDBSCAN, and OPTICS. The Silhouette Score s(i) for each point i was calculated as follows:$$s\left(i\right)=\frac{b\left(i\right)-a(i)}{max\;(a\left(i\right),b\left(i\right))}$$
where a(i) is the average intracluster distance, and b(i) is the minimum average inter-cluster distance [116]. Therefore, dimensionality reduction/machine learning tools can handle these large datasets, helping identify the relationships between omics data and clinical outcomes. By applying dimensionality reduction and unsupervised clustering algorithms and techniques, researchers can identify novel biomarkers and molecular signatures that can improve the accuracy of MSA diagnosis.

While these analytical approaches offer promising opportunities for advancing MSA diagnosis, this study has several limitations owing to the restricted sample size available for analysis. Firstly, since the most definitive confirmation of MSA is obtained through autopsies, patients were diagnosed based on clinical criteria widely recognized as defining features of MSA. Therefore, as this study primarily utilized CSF samples, pathological confirmation of MSA was not possible. Secondly, our study integrates datasets from various research groups with differing methodologies. Consequently, ensuring a strict control over these variations such as cohort size and FDR values and q values was not feasible. Additionally, the quality and consistency of samples were not standardized independently as they were obtained from contemporary reports. To overcome these limitations, a large and diverse population-based study on MSA is required in addition to more rigorous control over cohort size, sample consistency, FDR values for altered levels of molecules and use of more omics datasets such as microRNAs. High-throughput technologies and machine learning strategies should be used to identify pathophysiological patterns and multidimensional markers with a high affinity for MSA.

## 8. Summary

This review highlights the potential of metabolomic and proteomic analyses in advancing our understanding of MSA. Altered metabolic and proteomic profiles in the CSF of patients with MSA have been observed using literature-based networks and bioinformatics. We analysed the altered metabolomic and proteomic data in the CSF of patients with MSA to provide a more comprehensive perspective. By integrating omics approaches and utilising dimensionality reduction/machine learning tools, researchers can improve the identification of biomarkers and identify new therapeutic targets. Such advancements will not only enhance the diagnostic precision for MSA but also provide critical insights into the underlying mechanisms of MSA.

## Figures and Tables

**Figure 1 cells-14-00265-f001:**
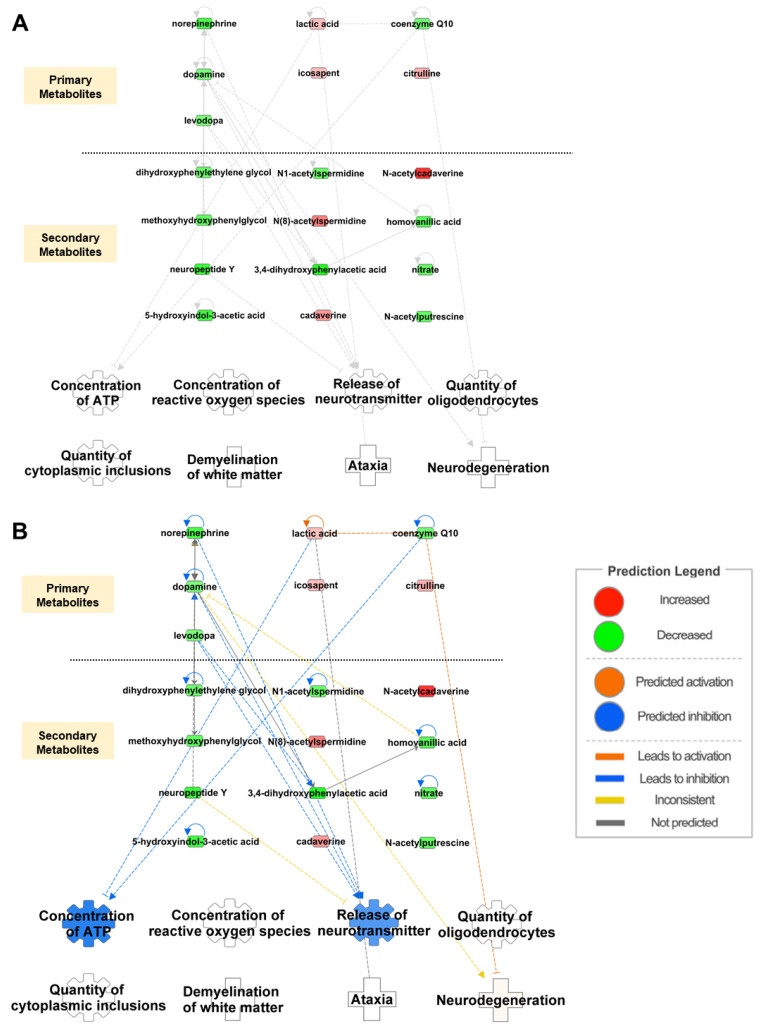
Biological functions and disease-related metabolomic networks from IPA. (**A**) Network of metabolites and their associated functions in the CSF of patients with MSA. (**B**) Prediction of the metabolomic network based on the altered metabolite levels. Red and green indicate increased and decreased metabolite levels, respectively, in the CSF of the patients. Orange and blue represent the activation and inhibition of functions and diseases, respectively. The solid and dotted lines represent direct and indirect relationships, respectively. The intensity of the predicted colour in the network is not a reflection of the severity of the disease and function itself but is related to the number of molecules available in the network to make the prediction.

**Figure 2 cells-14-00265-f002:**
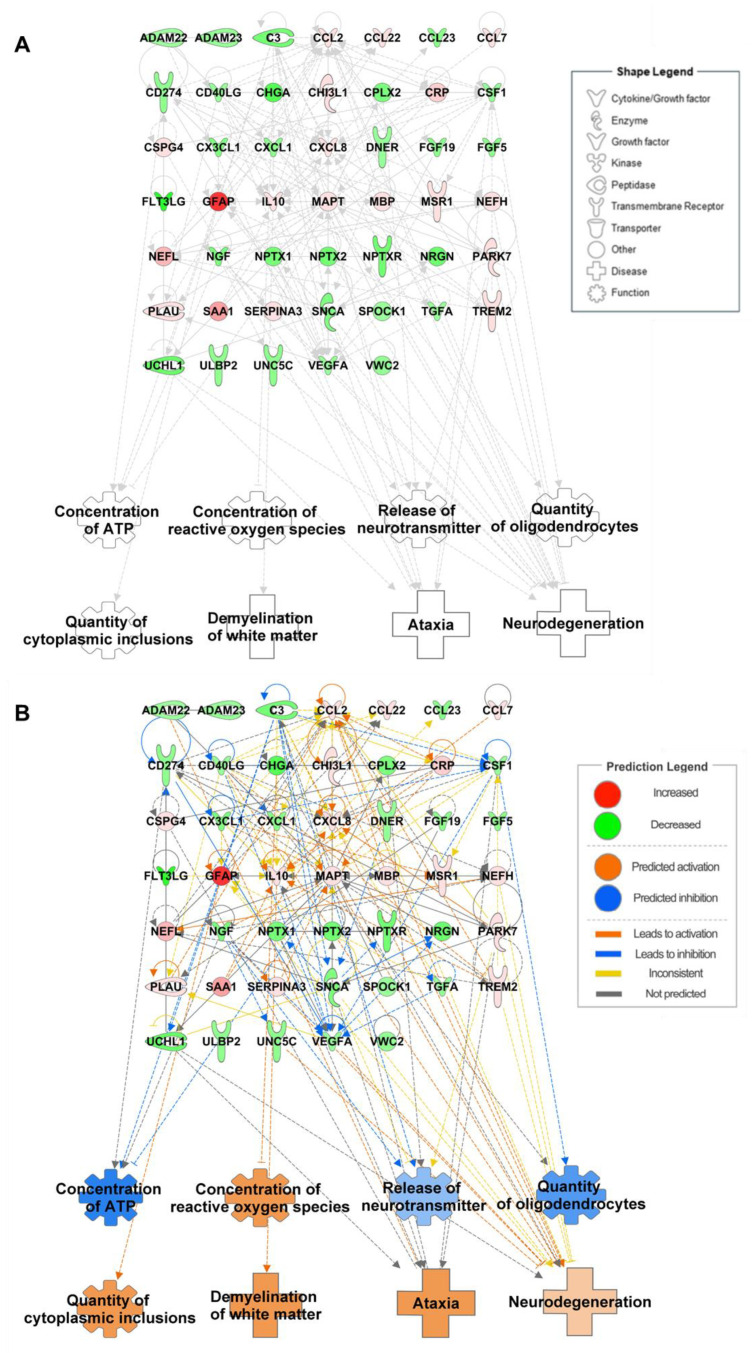
Biological functions and disease-related proteomic networks from IPA. (**A**) Network of proteins and their associated functions and diseases in the CSF of MSA patients. (**B**) Prediction of the proteomic network using altered metabolite levels. Red and green indicate increased and decreased levels of metabolites in the CSF of patients with MSA, respectively. Orange and blue represent the activation and inhibition of functions and diseases, respectively. Solid and dotted lines represent direct and indirect relationships, respectively.

**Figure 3 cells-14-00265-f003:**
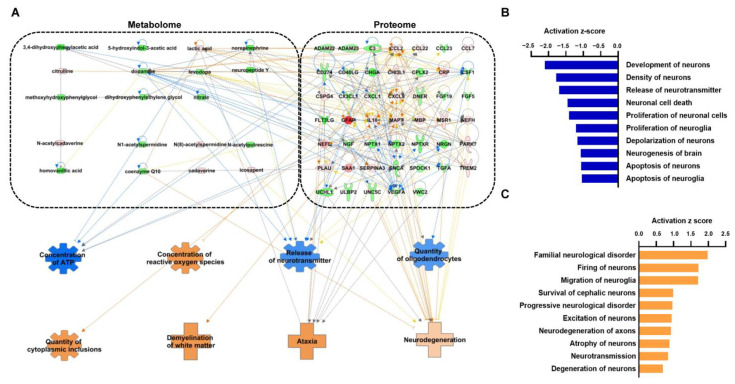
Biological functions and disease-related integrated networks from IPA. (**A**) Prediction of the integrated network based on the altered levels of metabolites and proteins in CSF of MSA patients. (**B**) Top 10 neurologically inhibited disease and biological functions according to z-score. (**C**) Top 10 neurologically activated disease and biological functions according to z-score.

**Figure 4 cells-14-00265-f004:**
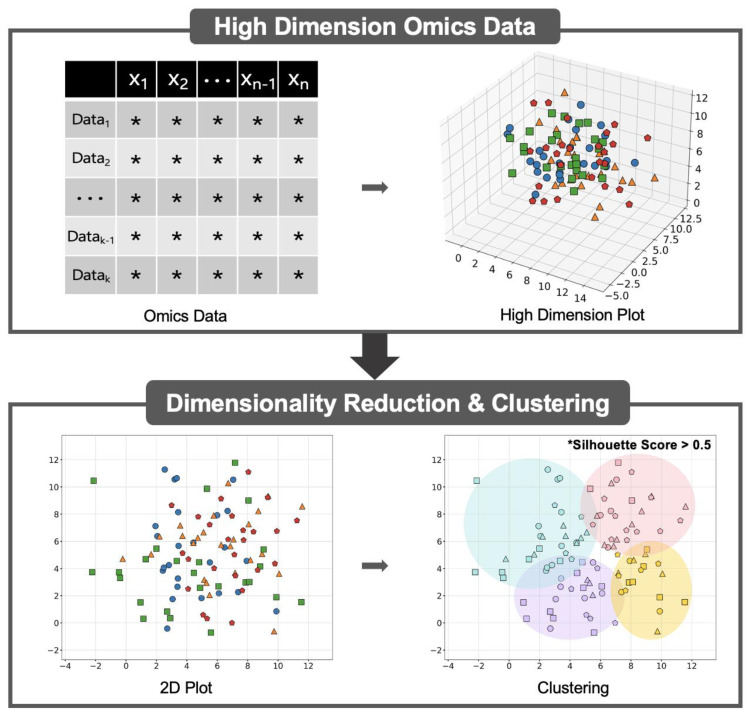
Workflow of high dimensional omics data analysis using dimensionality reduction and clustering. High dimensional omics data were first reduced to two dimensions using dimensionality-reduction techniques. Clustering methods were then applied to identify and trim outliers. In the omics data table, “*” represents randomly generated numerical values for illustration purposes. Different coloured symbols represent different types of omics data, while the colours in the clustering plot indicate distinct clusters identified through the analysis.

**Table 1 cells-14-00265-t001:** List of altered metabolites in CSF samples of MSA patients.

Metabolite	IPA Name	Expression Level	Fold Change	Analysis Method ^a^	*p* Value	References
	Primary Metabolites					
3,4-dihydroxyphenylalanine	Levodopa	Decrease	0.77	LC	<0.0001	[51]
Dopamine	Dopamine	Decrease	0.77	LC	<0.05	[51]
L-citrulline	Citrulline	Increase	1.31	HPLC	<0.05	[56]
Lactic Acid	Lactic acid	Increase	1.05	Amino acid analyser	0.04	[34]
Norepinephrine	Norepinephrine	Decrease	0.56	HPLC	<0.005	[57]
Eicosapentaenoic acid	Icosapent	Increase	1.08	GC-MS	0.006	[11]
Coenzyme Q10	Coenzyme Q10	Decrease	0.73	ELISA	0.036	[58]
	Secondary Metabolites					
*N*^1^-acetylcadaverine	N-acetylcadaverine	Increase	3.36	GC-MS	<0.001	[12]
*N*^1^-acetylspermidine	N1-acetylspermidine	Decrease	0.68	GC-MS	<0.001	[12]
*N*^8^-acetylsperimidine	N(8)-acetylspermidine	Increase	2.08	GC-MS	<0.005	[12]
*N*^1^-acetylputrescine	N-acetylputrescine	Decrease	0.66	GC-MS	<0.001	[12]
Cadaverine	Cadaverine	Increase	1.70	GC-MS	<0.007	[12]
Homovanillic acid	Homovanillic acid	Decrease	0.61	HPLC	0.003	[34]
Neuropeptide Y	Neuropeptide Y	Decrease	0.51	NPY-ir Assay	<0.01	[57]
3,4-dihydroxyphenylglycol	Dihydroxyphenylethylene glycol	Decrease	0.70	LC	<0.0001	[51]
3-methoxy-4-hydroxyphenylglycol	Methoxyhydroxyphenylglycol	Decrease	0.64	HPLC	<0.05	[57]
5-hydroxyindoleacetic acid	5-hydroxyindole-3- acetic acid	Decrease	0.54	HPLC	<0.0001	[59]
Nitrate	Nitrate	Decrease	0.66	ELISA	0.01	[60]
3,4-dihydroxyphenylacetic acid	3,4-dihydroxyphenylacetic acid	Decrease	0.46	LC	<0.0001	[51]

^a^ Abbreviations: LC, liquid chromatography; HPLC, high-performance liquid chromatography; GC-MS, gas chromatography with mass spectrometry; ELISA, enzyme-linked immunosorbent assay; NPY-ir, neuropeptide Y-immunoreactive.

**Table 2 cells-14-00265-t002:** List of altered protein levels in cerebrospinal fluid of MSA patients.

Protein ^a^	IPA Name	Accession Number ^b^	Expression Level	Fold Change	Analysis Method ^c^	*p* Value	References
NFL	NEFL	P07196	Increase	6.51	SIMOA	<0.0001	[80]
GFAP	GFAP	P14136	Increase	1.88	SIMOA	<0.01	[80]
SYUA	SNCA	P37840	Decrease	0.75	Immunoassay	<0.05	[81]
TAU	MAPT	P10636	Increase	1.48	Innotest hTau assay	<0.0001	[10]
MBP	MBP	P02686	Increase	1.6	ELISA	<0.001	[82]
CH3L1	CHI3L1	P36222	Increase	1.54	Immunoassay	<0.05	[81]
CCL2	CCL2	P13500	Increase	1.27	Immunoassay	<0.05	[81]
CRP	CRP	P02741	Increase	4.53	Biomarkers Kit	<0.05	[83]
SAA1	SAA1	P0DJI8	Increase	8.66	Biomarkers Kit	<0.001	[83]
IL8	CXCL8	P10145	Increase	1.21	Biomarkers kit	<0.05	[83]
UCHL1	UCHL1	P09936	Decrease	0.63	Sandwich ELISA	<0.05	[84]
PARK7	PARK7	Q99497	Increase	1.69	Sandwich ELISA	<0.001	[85]
X3CL1	CX3CL1	P78423	Decrease	0.75	Proximity Extension Assay	<0.05	[86]
CCL7	CCL7	P80098	Increase	1.22	Multiplex Assay	0.00001	[46]
IL10	IL10	P22301	Increase	1.32	Multiplex Assay	0.00001	[46]
CCL22	CCL22	O00626	Increase	1.41	Multiplex Assay	0.00001	[46]
CO3	C3	P01024	Decrease	0.69	Bead-based Luminex Assay	<0.05	[87]
FLT3L	FLT3LG	P49771	Decrease	0.45	Luminex Assay	<0.001	[88]
FGF19	FGF19	O95750	Decrease	0.82	Proximity Extension Assay	<0.05	[86]
CD40L	CD40LG	P29965	Decrease	0.92	Proximity Extension Assay	<0.05	[86]
PD1L1	CD274	Q9NZQ7	Decrease	0.85	Proximity Extension Assay	<0.05	[86]
TGFA	TGFA	P01135	Decrease	0.91	Proximity Extension Assay	<0.05	[86]
CSF1	CSF1	P09603	Decrease	0.91	Proximity Extension Assay	<0.05	[86]
UROK	PLAU	P00749	Increase	1.13	Proximity Extension Assay	<0.05	[86]
VEGFA	VEGFA	P15692	Decrease	0.92	Proximity Extension Assay	<0.05	[86]
CCL23	CCL23	P55773	Decrease	0.80	Proximity Extension Assay	<0.05	[86]
GROA	CXCL1	P09341	Decrease	0.81	Proximity Extension Assay	<0.05	[86]
DNER	DNER	Q8NFT8	Decrease	0.98	Proximity Extension Assay	<0.05	[86]
NGF	NGF	P01138	Decrease	0.72	Proximity Extension Assay	<0.05	[86]
NEUG	NRGN	Q92686	Decrease	0.65	ELISA	<0.001	[89]
NFH	NEFH	P12036	Increase	2.50	ELISA	<0.001	[89]
CMGA	CHGA	P10645	Decrease	0.60	Sandwich ELISA	0.014	[90]
FG5	FGF5	P12034	Decrease	0.83	Proximity Extension Assay	<0.05	[91]
MSRE	MSR1	P21757	Increase	1.27	Proximity Extension Assay	<0.05	[91]
VWC2	VWC2	Q2TAL6	Decrease	0.89	Proximity Extension Assay	<0.05	[91]
ADA22	ADAM22	Q9P0K1	Decrease	0.97	Proximity Extension Assay	<0.05	[91]
UNC5C	UNC5C	O95185	Decrease	0.84	Proximity Extension Assay	<0.05	[91]
ADA23	ADAM23	O75077	Decrease	0.91	Proximity Extension Assay	<0.05	[91]
T1CN1	SPOCK1	Q08629	Decrease	0.95	Proximity Extension Assay	<0.05	[91]
ULBP2	ULBP2	Q9BZM5	Decrease	0.86	Proximity Extension Assay	<0.05	[91]
TREM2	TREM2	Q9NZC2	Increase	1.75	Multiplex Assay	<0.001	[92]
CSPG4	CSPG4	Q6UVK1	Increase	1.22	Biomarker Assay	0.0234	[45]
NPTX1	NPTX1	Q15818	Decrease	0.71	LC-MS	<0.01	[93]
NPTX2	NPTX2	P47972	Decrease	0.68	LC-MS	<0.01	[93]
NPTXR	NPTXR	O95502	Decrease	0.68	LC-MS	<0.01	[93]
CPLX2	CPLX2	Q6PUV4	Decrease	0.73	LC-MS	<0.01	[93]
AACT	SERPINA3	P01011	Increase	1.29	Immunoturbidimetry	<0.05	[94]

Abbreviations: ^a^ Protein symbol indicates the human protein assigned by UniProtKB. ^b^ Accession number indicates the unique identifier of the protein in UniProtKB. ^c^ Abbreviations: SIMOA, single molecule array; ELISA, enzyme-linked immunosorbent assay; LC-MS/MS, liquid chromatography–mass spectrometry; SP assay, substance P assay; Co-IP with MS, co-immunoprecipitation with mass spectrometry.

## Data Availability

No new data were created or analyzed in this study.

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
