# Peer review of "Integrative Analysis of Metabolome and Proteome in the Cerebrospinal Fluid of Patients with Multiple System Atrophy"

_cells, 2025, doi:10.3390/cells14040265_

Round 1

Reviewer 1 Report

Comments and Suggestions for Authors

Manuscript cells-3427178

Nimisha Pradeep George et al.

This manuscript, “Integrative Analysis of Metabolome and Proteome in the Cerebrospinal Fluid of Patients with Multiple System Atrophy.,” is written by Nimisha Pradeep George et al.

Multiple System Atrophy (MSA) is a fatal progressive neurodegenerative disease that manifests with autonomic dysfunction, Parkinsonism, and cerebellar ataxia. MSA is classified as a disease characterized by the accumulation of α-synuclein, similar to Parkinson's disease; however, it generally progresses more rapidly. The identification of biomarkers that can differentiate between these conditions is of critical importance. This manuscript is a potentially important study that integrates cerebrospinal fluid metabolomics and proteomics from MSA patients, based on a literature review, to identify biomarkers rooted in the pathophysiology of MSA.

However, the reviewer has some concerns. Essential information, such as the search algorithm used for the literature review, the number of patients included in the reviewed studies, and their clinical characteristics, is not provided. Without these details, it is difficult to assess the reliability of the metabolite and protein changes extracted by the authors. Consequently, there is a potential risk of bias in the selection of metabolites and proteins. Since this manuscript conducts IPA analysis based on information from reviewed literature, it has the characteristics of a systematic review. Therefore, the analysis should adhere to methodologies consistent with systematic review principles.

This reviewer’s concerns are listed below.

Major issues:

1.        In Section 4, the authors have described the search terms used for reviewing the literature in PubMed and Google Scholar. However, they have not provided key details, such as the number of papers identified, the criteria for including or excluding studies, and the algorithms used for selection. This raise concerns that the selection of studies may have been arbitrary and could introduce potential bias.

2.        In Section 4, Table 1, studies with comprehensive analyses using LC or GC-MS are listed alongside studies measuring single metabolites using ELISA. While both approaches identify significant metabolites using p-values, shouldn't comprehensive analyses account for multiple comparisons by applying appropriate corrections to the p-values? If possible, FDR or q-value should be accompanied.

3.        Is it possible that the references cited by the authors are included in the database used by the IPA analysis? In other words, if the cited references are already part of the IPA database, could this create a bias where neuron-related pathways are naturally enriched?

4.        In the lower part of Figure 1A, eight molecular functions are listed. How were these functions selected? Do they represent the top eight functions identified through statistical analysis in the IPA analysis, or are they functions that the authors subjectively selected? If the latter, there is a potential for bias in their selection.

5.        In Section 5, the authors should similarly provide the criteria for selecting studies from the search results. Additionally, information such as the number of patients included in the selected studies, fold change data, and other relevant details should be presented. Furthermore, issues related to multiple comparisons should also be addressed, as in previous sections.

6.        Throughout the study, a decrease in ATP production is predicted. However, was ATP itself detected in the metabolome analysis? While ATP can sometimes be detected using GC-MS, it would be important to verify whether ATP was identified in the cited references. Clarifying this would strengthen the validity of the conclusion.

Minor issues:

1.        In Section 4, the authors need to clearly define the terms “primary metabolite” and “secondary metabolite.”

2.        In Table 1 of Section 4, p-values are reported, but fold change is also critical information and should be included. Additionally, is fold change not considered in the IPA analysis?

3.        In Figure 1 of Section 4, the legend appears to be incomplete. For example, the combination of arrows and colors makes it difficult to determine whether a process is being promoted or inhibited. Providing clearer indications in the legend would improve interpretability.

4.        In Section 5, Figure 2 illustrates the relationship with molecular functions similar to those in Figure 1. Is this an indication that the same functions were enriched coincidentally? Clarification on this point would strengthen the interpretation of the results.

5.        On Page 8, Lines 256-257, there is a mention of the correlation between CSF-1 and the number of oligodendrocytes. How was the number of oligodendrocytes measured? Providing details about the methodology used for this measurement would clarify the validity of the correlation. Does this citation refer to a study conducted in mice?

6.        In Section 6, Figure 3 C and D, is the exclusive enrichment of neuron-related terms in the IPA analysis due to any specific preprocessing steps, or does the enrichment of neuron-specific pathways naturally result from the metabolites and proteins extracted for IPA analysis?

Author Response

Answers for Reviewer 1

Thank you for your insightful suggestions and positive feedback. We appreciate both the critical and encouraging comments. We have carefully addressed each point and revised the manuscript accordingly. Below, we provide a detailed point-by-point response, with your comments in black, our responses in blue, and manuscript changes highlighted in red. The following is our response summary.

Major Revision

Comment 1 - In Section 4, the authors have described the search terms used for reviewing the literature in PubMed and Google Scholar. However, they have not provided key details, such as the number of papers identified, the criteria for including or excluding studies, and the algorithms used for selection. This raise concerns that the selection of studies may have been arbitrary and could introduce potential bias.

Response 1- Thank you for pointing out this oversight. Based on your suggestion, we have revised our statement to include key details regarding the selection of studies. By adhering to these predefined criteria, we ensured that the dataset curated for IPA analysis was based on robust and reproducible data, minimizing any risk of arbitrary selection or potential bias. This careful approach allowed us to focus on meaningful metabolite changes specifically associated with MSA, enhancing the validity of our review and subsequent analyses. To our knowledge, this is a first attempt at in silico prediction using both metabolomic and proteomic data. Future studies will be executed using more data and therefore a larger dataset.  

Revised - Lines 169-183

For the data acquisition process, we conducted a systematic literature search using the terms “biomarkers,” “metabolite” or “metabolome,” “MSA” or “multiple system atrophy,” and “CSF” or “cerebrospinal fluid,” in PubMed and Google Scholar. These search terms were applied in various combinations to ensure comprehensive coverage of the available literature. Through this process, we identified 16 studies reporting metabolite biomarkers in MSA patients compared to healthy controls using CSF samples. To ensure a rigorous and unbiased selection of studies, we adhered to a set of inclusion and exclusion criteria. The inclusion criteria were studies reporting metabolites with statistically significant fold changes between MSA patients and healthy controls, studies that provided explicit fold-change data or other quantifiable measures for comparison and comparisons restricted to MSA patients and normal healthy controls. Exclusion criteria for this study excluded metabolites without reported fold-change values or statistical significance, studies including comparisons with disease groups other than MSA or controls and reports where metabolite changes were not statistically validated.

Comment 2 - In Section 4, Table 1, studies with comprehensive analyses using LC or GC-MS are listed alongside studies measuring single metabolites using ELISA. While both approaches identify significant metabolites using p-values, shouldn't comprehensive analyses account for multiple comparisons by applying appropriate corrections to the p-values? If possible, FDR or q-value should be accompanied.

Response 2- We appreciate your insightful comment and agree with your observation. Since this review compiles datasets from multiple studies with varying methodologies, strict control over these variations was not feasible. However, we minimized bias by following predefined inclusion and exclusion criteria. Recognizing the importance of statistical variability, we acknowledge your suggestion to incorporate more rigorous methods like as false discovery rate (FDR) corrections or q-values. To address this, we have highlighted it as a key limitation. In future studies, we will implement stricter dataset selection criteria, including better control over cohort size.

Revised – Lines 421 - 423

Secondly, our study integrates datasets from various research groups with differing methodologies. Consequently, ensuring a strict control over these variations such as cohort size and FDR values and q values was not feasible

Comment 3 - Is it possible that the references cited by the authors are included in the database used by the IPA analysis? In other words, if the cited references are already part of the IPA database, could this create a bias where neuron-related pathways are naturally enriched?

Response 3 - The references we cited may overlap with the databases used by IPA, which integrates data from extensive peer-reviewed literature, including pathways and gene functions. However, as IPA predictions are based solely on our provided dataset, constructed using fold change values and statistically significant changes, it reduces the possibility of any bias. If neuron-related pathways are enriched, it highlights the key role these molecules play in MSA pathophysiology.

Revised – Lines 343 - 346

Since the IPA program makes predictions based on a curated dataset, it helps minimize potential bias. The enrichment of neuron-related pathways underscores the essential role of these biomarkers in the pathophysiology of MSA.

Comment 4 - In the lower part of Figure 1A, eight molecular functions are listed. How were these functions selected? Do they represent the top eight functions identified through statistical analysis in the IPA analysis, or are they functions that the authors subjectively selected? If the latter, there is a potential for bias in their selection.

Response 4 - The eight molecular functions listed (ATP concentration, reactive oxygen species, neurotransmitter release, oligodendrocyte quantity, cytoplasmic inclusions, demyelination, ataxia, and neurodegeneration) are hallmark features of MSA, as outlined in Section 2 of our manuscript and supported by multiple sources. Since IPA utilizes biological functions, diseases, and pathways to make predictions, it's essential to focus on those most closely related to MSA or typical characteristics exhibited in MSA pathology. The predictions, based on fold change from the curated dataset, ensure robust and unbiased results.

Revised – Lines 188-194

From the broad range of functions and diseases available for prediction in the IPA program, five functions and three diseases representing key MSA hallmarks were selected. Since the predictions are based solely on a meticulously curated dataset, incorporating fold changes and statistical significance of molecular expression alterations in MSA patients compared to healthy controls, this approach minimizes any potential bias that might arise from subjective selection of functions and diseases.

Comment 5 - In Section 5, the authors should similarly provide the criteria for selecting studies from the search results. Additionally, information such as the number of patients included in the selected studies, fold change data, and other relevant details should be presented. Furthermore, issues related to multiple comparisons should also be addressed, as in previous sections.

Response 5 – We appreciate your feedback and have made the necessary revisions by adding supplemental tables for both metabolomic and proteomic data. We added fold change in the main tables and provide supplementary tables with sample size information.

Revised – Lines 184 - 187

Of these, only 19 metabolites showed a statistically significant difference in their metabolic profiles compared to healthy controls; therefore, only these molecules were uploaded to the IPA program for analysis and network construction (Table 1, Table S1, Figure 1A).

Revised – Lines 274 -276

Among them, only 47 proteins showed statistically significant changes in their expression levels in patients with MSA compared with healthy controls; therefore, only these proteins were included in the dataset for the IPA analysis (Table 2, Table S3).

Comment 6 - Throughout the study, a decrease in ATP production is predicted. However, was ATP itself detected in the metabolome analysis? While ATP can sometimes be detected using GC-MS, it is important to verify whether ATP was identified in the cited references. Clarifying this would strengthen the validity of the conclusion.

Response 6 – Although a decrease in ATP production is predicted, ATP levels itself were not detected in the datasets used to curate the metabolomic dataset. ATP is a critical metabolite that reflects well mitochondrial function or dysfunction. While curating the dataset, we did not differentiate between studies that analysed ATP itself with those that did not in their analysis of metabolites. The biological function “concentration of ATP” was not chosen as it was detected using GC-MS in the cited references but rather because it is a critical component in the functioning of neurons, mitochondrial function and ultimately neurodegeneration. The prediction itself was solely based on the dataset curated using fold change. Our in silico predictions aligns with these reports as both ATP level and concentration of ATP are predicted to be decreased as shown below. However, this is a profound comment and we have made the following changes in the manuscript.

Revised – Lines 209 -210

This aligns with our in silico prediction that shows increased lactic acid levels directly impact ATP levels itself (Figure S2).

Figure S2 see attached PDF.

Minor issues:

  1. In Section 4, the authors need to clearly define the terms “primary metabolite” and “secondary metabolite.”

Response 1 – Thank you for bringing this to our attention. Acknowledging your advice on defining the terms “primary metabolite” and “secondary metabolite”, we have added our criteria for defining primary and secondary metabolite to our manuscript.

Revision – Lines 155-160

Primary metabolites such as amino acids and lipids are critical for energy production, neurotransmitter synthesis and structural integrity for neurons. They are produced through fundamental metabolic pathways such as TCA cycle. Secondary metabolites such as neuroactive peptides serve specialized roles in neuroprotection, signaling, and adaptation to stress and are derived from primary metabolites or complex pathways.

  1. In Table 1 of Section 4, p-values are reported, but fold change is also critical information and should be included. Additionally, is fold change not considered in the IPA analysis?

Response 2 – We fully concur with your suggestion. Fold change is critical in  creating datasets especially in making predictions using IPA program. To rectify this lapse, we have added the values in main tables ( Table 1 for metabolites and Table 3 for proteins).

Revised – Lines 184 - 187

Of these, only 19 metabolites showed a statistically significant difference in their metabolic profiles compared to healthy controls; therefore, only these molecules were uploaded to the IPA program for analysis and network construction (Table 1, Table S1, Figure 1A).

Revised – Lines 274 -276

However, only 47 proteins showed statistically significant changes in their expression levels in patients with MSA compared with healthy controls; therefore, only these proteins were included in the dataset for the IPA analysis (Table 2, Table S3).

  1. In Figure 1 of Section 4, the legend appears to be incomplete. For example, the combination of arrows and colors makes it difficult to determine whether a process is being promoted or inhibited. Providing clearer indications in the legend would improve interpretability.

Response 3 – You’ve brought up a critical point for our manuscript and we are in agreement with this feedback. Therefore, as per your suggestion to provide clearer indications in the legend we have revised the legend in Figure 1.

Revised -Figure 1 with a clearer prediction legend

  1. In Section 5, Figure 2 illustrates the relationship with molecular functions similar to those in Figure 1. Is this an indication that the same functions were enriched coincidentally? Clarification on this point would strengthen the interpretation of the results.

Response 4 – Thank you for your insightful comment. We agree that Figure 2 shows similar molecular functions to Figure 1. The IPA program uses a wide range of biological functions, diseases, and pathways for predictions, many of which are well-documented as hallmarks of MSA. This overlap strengthens the hypothesis that these functions are central to MSA pathophysiology. Additionally, we applied multiple comparison corrections during functional enrichment analysis to minimize random overlaps and ensure the validity of our findings.

Revised – Lines 188 - 194

From the broad range of functions and diseases available for prediction in the IPA program, five functions and three diseases representing key MSA hallmarks were selected. Since the predictions are based solely on a meticulously curated dataset, incorporating fold changes and statistical significance of molecular expression alterations in MSA patients compared to healthy controls, this approach minimizes any potential bias that might arise from subjective selection of functions and diseases.

  1. On Page 8, Lines 256-257, there is a mention of the correlation between CSF-1 and the number of oligodendrocytes. How was the number of oligodendrocytes measured? Providing details about the methodology used for this measurement would clarify the validity of the correlation. Does this citation refer to a study conducted in mice?

Response – The literature source for lines 256 -257 mentioning correlation between CSF-1 and the number of oligodendrocytes used both ex vivo and in vitro methods to determine this relationship (https://doi.org/10.1016/j.expneurol.2019.04.011). Oligodendrocyte cell number in cerebellar explants of mice and in purified primary murine cultures were quantified by immunostaining and cell marker specific staining respectively. Taking your feedback into consideration, the statement has been revised to read as follows.

Revised - Line 298 - 303

Liu et al. performed both ex vivo and in vivo experiments on cerebellar explants from mice and in purified primary murine culture respectively to study the effect of inhibition of CSF-1 on oligodendrocyte precursor cells (OPC). They quantified the number of OPCs using immunostaining methods and established that inhibiting CSF-1 directly impaired OPC viability in vitro and caused a reduction in OPC numbers ex vivo and in vivo.

  1. In Section 6, Figure 3 C and D, is the exclusive enrichment of neuron-related terms in the IPA analysis due to any specific preprocessing steps, or does the enrichment of neuron-specific pathways naturally result from the metabolites and proteins extracted for IPA analysis?

Response 6 – Thank you for highlighting this point. Figure 3C and D (now 3B and C) show the top 10 inhibited and activated diseases and biological functions based on the activation z-score. The IPA program includes both neurological and non-neurological functions, but since our study focuses on MSA and neurodegeneration, we filtered for the top 10 relevant neurological functions. To clarify this, we will revise the manuscript accordingly.

Revised – Line 360-362

The top 10 neurological diseases and biological functions according to the activation z-score predicted to be inhibited (Figure 3B, Table S5) and activated (Figure 3C, Table S6) are shown.

Reviewer 2 Report

Comments and Suggestions for Authors

In the article from George NP et al., the authors review and analyze the current molecular understanding of CSF in Multiple System Atrophy patients. The article recognizes both the significant progress made in molecular profiling of CSF, with assembly of metabolomes and proteomes, as well as the challenges faced in characterizing a disease that is currently only definitively diagnosed at the post-mortem stage. The authors assemble in silico network of metabolites and proteins that have been found dysregulated across different studies, and then integrate these network to identify the top pathways of interest in MSA.  The authors approach is well conceptualized, however, the execution could be improved. For example, it is not clear if all the relevant studies were marked by the authors (see below). Generally, the authors could improve the clarity when describing new results/analyses, and theoretical computational biology approaches that have been broadly useful in the field of neurodegenerative diseases.

Major issues

1.       There is an open question whether the authors’ literature curation of studies was sufficiently comprehensive. For the proteomics studies, the authors did not report using “proteomics” as a keyword, only “protein” (line 228-229). I found two potential proteomics studies that included. MSA patients, https://pubmed.ncbi.nlm.nih.gov/39164482/ and https://pubmed.ncbi.nlm.nih.gov/35657417/ that do not appear to be cited in the current manuscript.  For these studies, if they haven’t already, I suggest the authors review these studies for eligibility or provide rationale while they could not be included. The authors may also want to include the keyword proteomics to see if additional studies are identified.

2.       Given the value of multi-omics to provide complementary readouts of the same disease state (as the authors have stated), the authors should consider including transcriptomics analyses in their computational analysis.  Datasets from Piras et al., (https://www.biorxiv.org/content/10.1101/2020.02.11.944306v1)l, and Bettencourt et al. (https://pubmed.ncbi.nlm.nih.gov/32410669/) appear to be fine candidates, but a comprehensive survey of the literature should be performed.

3.       The reason for presenting Figure 4 concepts of multi-omics integration and dimensionality reduction in section 6 is unclear.  Figure 3 is a concrete result obtained by the authors. However, unless I am missing a key point, Fig 4 appears to be theoretical? If so, perhaps the authors could include this as a separate “Future directions” section, how improvements in computational and data integration strategies will be useful as more omics datasets are studies for MSA.

Minor issues

·        This statement in the abstract is confusing and may need revision: “Recognizing that metabolomics has a limited capacity for capturing small molecules, we also integrated proteomic data to enhance the overall comprehensiveness of the study.”

·        These sentences seem contradictory as the first suggests that MSA is “curable”, but the second sentence refutes that. “As MSA progresses more aggressively than other synucleinopathies and has a limited response to symptomatic treatment, early and accurate diagnosis is critical for curing patients [9]. Unfortunately, no current disease-modifying therapies or accurate biomarkers are available for MSA, and symptomatic management remains limited.

·        The authors should provide a supplemental table that lists the studies used for the metanalysis of (A) metabolites and (B) proteins.

·        The sentence describing metabolomes and proteomes (53-55) described metabolomes as “quantitative” but was written in a way that implied proteomes were not also quantitative.  The authors should clarify this statement.

·        The authors cite their previous studies that the levels of FFA and polyamines (PA) in CSF were distinguishing features between PD and MSA patients.  I may be missing the distinction, but the study (ref 11) for FFA did not show a difference for EPA between PD and MSA.  Thus only the polyamines could serve as a distinguishing biomarker.  Could the authors clarify ref 11 results, or modify the text to accuracy reflect the prior work.

·        The sentence starting “These multiomics approaches…” on line 74, is true, but does not seem relevant to the manuscript as single-cell proteomics was not considered/has not been performed for MSA.

·        In Fig 1A and 2A, there are some functions not connected to specific metabolites. How were these selected for addition to the figure if they are not connected to the target molecules?

·        In Fig 3, the network in B that represents the predicted functionality is the main result of interest. I suggests panel A be included the supplement to allow panel B to be larger and more readable.

Author Response

Answers to Reviewer 2

Thank you for your extremely insightful and valuable feedback. We have carefully considered and addressed each aspect, revising the manuscript accordingly. Below, we present a structured response, with your comments in black, our replies in blue, and the changes made to the manuscript highlighted in red.

Major Revision

Comments 1: There is an open question whether the authors’ literature curation of studies was sufficiently comprehensive. For the proteomics studies, the authors did not report using “proteomics” as a keyword, only “protein” (line 228-229). I found two potential proteomics studies that included. MSA patients, https://pubmed.ncbi.nlm.nih.gov/39164482/ and https://pubmed.ncbi.nlm.nih.gov/35657417/ that do not appear to be cited in the current manuscript.  For these studies, if they haven’t already, I suggest the authors review these studies for eligibility or provide rationale while they could not be included. The authors may also want to include the keyword proteomics to see if additional studies are identified.

Response 1 - Thank you for your input. We came across these publications but excluded them because the CSF samples were fractionated into exosomes, microvesicles, and soluble proteins, whereas most studies did not specify such fractionation in their methods. To maintain data consistency, we opted to exclude studies with explicit CSF fractionation. Additionally, while we used "proteomics" alongside "MSA" and "CSF" in our search, we acknowledge an oversight in keyword selection and will revise accordingly.

Revised – Line 258 - 273

For the data acquisition processes, the search terms used in different combinations were “biomarkers”, “protein,” “proteomics”, “MSA or multiple system atrophy,” and “CSF or cerebrospinal fluid” in PubMed and Google Scholar. These keywords were used in different sequences to obtain a thorough exposure of the available literature. Through this process, we identified 31 studies reporting protein biomarkers in MSA patients compared to healthy controls using CSF samples. To ensure a thorough and impartial selection of studies, we followed predefined inclusion and exclusion criteria. The inclusion criteria were studies reporting metabolites with statistically significant fold changes between MSA patients and healthy controls, studies that provided explicit fold-change data or other quantifiable measures for comparison, studies employing whole CSF samples for protein profiling, and comparisons restricted to MSA patients and normal healthy controls. Exclusion criteria for this study excluded metabolites without reported fold-change values or statistical significance, studies including comparisons with disease groups other than MSA or controls, studies using fractionated CSF samples to study protein concertation in individual fractions, and reports where metabolite changes were not statistically validated.

Comment 2: Given the value of multi-omics to provide complementary readouts of the same disease state (as the authors have stated), the authors should consider including transcriptomics analyses in their computational analysis.  Datasets from Piras et al., (https://www.biorxiv.org/content/10.1101/2020.02.11.944306v1)l, and Bettencourt et al. (https://pubmed.ncbi.nlm.nih.gov/32410669/) appear to be fine candidates, but a comprehensive survey of the literature should be performed.

Response 2 – We agree with your feedback that transcriptomic analysis enables a more comprehensive study into MSA pathology. The two papers that you brought to our attention use human brain tissue for gene profiling.

microRNAs (miRNAs) regulate post transcriptional gene expression and are stable in biofluids such as CSF making them excellent sources of biomarkers. However, there are several challenges that come with using miRNAs in a study like ours that sources dataset from various research groups and literature sources namely the normalization techniques that could possible affect their expression levels, difference in baseline mRNA due to age, severity of disease or any other factors, cohort size and so on. Acknowledging the importance of your comment , we address this in our limitations and revised the manuscript.

Revised – Lines 425 - 428

To overcome these limitations, a large and diverse population-based study on MSA is required in addition to more rigorous control over cohort size, sample consistency, FDR values for altered levels of molecules and use of other omics datasets such as microRNAs.

Comment 3 - The reason for presenting Figure 4 concepts of multi-omics integration and dimensionality reduction in section 6 is unclear.  Figure 3 is a concrete result obtained by the authors. However, unless I am missing a key point, Fig 4 appears to be theoretical? If so, perhaps the authors could include this as a separate “Future directions” section, how improvements in computational and data integration strategies will be useful as more omics datasets are studies for MSA.

Response 3 – Thank you for bringing this to our attention. Therefore, we have added a new section to incorporate improvements in computational and data integration strategies can be useful in studying omics datasets for MSA.

Revised – Line 379

  1. Future directions using machine learning

Minor Revision

Comment 1 -This statement in the abstract is confusing and may need revision: “Recognizing that metabolomics has a limited capacity for capturing small molecules, we also integrated proteomic data to enhance the overall comprehensiveness of the study.

Response 1- As per your comment that the statement in the abstract is confusing, we have revised the statement as follows:

Revised - Line 18-20

Acknowledging the limitations of metabolomics, we incorporated proteomic data to improve the overall comprehensiveness of the study.

Comment 2- These sentences seem contradictory as the first suggests that MSA is “curable”, but the second sentence refutes that. “As MSA progresses more aggressively than other synucleinopathies and has a limited response to symptomatic treatment, early and accurate diagnosis is critical for curing patients [9]. Unfortunately, no current disease-modifying therapies or accurate biomarkers are available for MSA, and symptomatic management remains limited.

Response 2 – We acknowledge your suggestion that the statements seem contradictory and therefore have revised our statement to read as follows:

Revised - Lines 38 - 40

As MSA progresses more aggressively than other synucleinopathies and has a limited response to symptomatic treatment, early and accurate diagnosis is critical for treatment and symptom management.

Comment 3 - The authors should provide a supplemental table that lists the studies used for the metanalysis of (A) metabolites and (B) proteins.

Response 3 – Thank you for you comment. As you pointed out, we have included fold change values in the main tables and provided supplemental tables (S1 and S3) with sample size for the metabolites and proteins respectively mentioned in Table 1 and 2 respectively.

Revised – Lines 184 - 187

Of these, only 19 metabolites showed a statistically significant difference in their metabolic profiles compared to healthy controls; therefore, only these molecules were uploaded to the IPA program for analysis and network construction (Table 1, Table S1, Figure 1A).

Revised – Lines 274 - 276

Among them, only 47 proteins showed statistically significant changes in their expression levels in patients with MSA compared with healthy controls; therefore, only these proteins were included in the dataset for the IPA analysis (Table 2, Table S3).

Comment 4 - The sentence describing metabolomes and proteomes (53-55) described metabolomes as “quantitative” but was written in a way that implied proteomes were not also quantitative.  The authors should clarify this statement.

Response 4- Your feedback is valid and taking it into consideration, we discarded the term quantitative for metabolome and the revised statement is as follows –

Revised – Lines 56-60

Because metabolomes can effectively showcase phenotypic traits, while proteomes provide a dynamic representation of cellular states and direct functional insight into CSF, there is a critical need to analyse changes in the metabolome and proteome across different body fluids, especially the CSF, which flows through the ventricular system of the brain.

Comment 5 - The authors cite their previous studies that the levels of FFA and polyamines (PA) in CSF were distinguishing features between PD and MSA patients.  I may be missing the distinction, but the study (ref 11) for FFA did not show a difference for EPA between PD and MSA.  Thus, only the polyamines could serve as a distinguishing biomarker.  Could the authors clarify ref 11 results or modify the text to accuracy reflect the prior work.

Response 5 -  We agree that there is no significant difference in EPA levels in CSF sample between PD and MSA patients. However, for our report we are only considering EPA levels in CSF samples between MSA and healthy controls. Taking your comment into consideration, we have made the following change to our manuscript.

Revised – Lines 50 -55

Elevated  levels of the free fatty acid eicosapentaenoic acid (EPA) were observed in the CSF of both MSA and PD patients compared to healthy controls. However, no significant difference was found between MSA and PD, emphasizing the need for further metabolomic studies to distinguish MSA from other synucleinopathies which is critical for implementing appropriate treatment strategies at the onset of each disease.

Comment 6 - The sentence starting “These multiomics approaches…” on line 74, is true, but does not seem relevant to the manuscript as single-cell proteomics was not considered/has not been performed for MSA.

Response 6 – We agree with your comment and as single cell proteomic was not considered/ has not been performed for MSA, we have revised the statement.

Revised – Lines 76-78

These multiomics approaches enable the capture of more complex biological interactions, allowing the identification of novel molecular signatures that reflect disease mechanisms.

 Comment 7 -  In Fig 1A and 2A, there are some functions not connected to specific metabolites. How were these selected for addition to the figure if they are not connected to the target molecules?

Response 7 – In Figures 1A and 1B, some functions are not linked to specific metabolites due to the limited number of metabolites available for study. These functions were selected as key hallmarks of MSA pathology, and the dataset was carefully curated based on fold change and statistical significance to minimize bias. To address this limitation, we integrated the metabolomic data with the proteomic network in Figure 3, enhancing the robustness of our predictions.

Revised – Lines 188 - 194

From the broad range of functions and diseases available for prediction in the IPA program, five functions and three diseases representing key MSA hallmarks were selected. Since the predictions are based solely on a meticulously curated dataset, incorporating fold changes and statistical significance of molecular expression alterations in MSA patients compared to healthy controls, this approach minimizes any potential bias that might arise from subjective selection of functions and diseases.

Comment 8 - In Fig 3, the network in B that represents the predicted functionality is the main result of interest. I suggests panel A be included the supplement to allow panel B to be larger and more readable.

Response 8 -  Thank you for your comment. We agree with your comment and have made the appropriate changes to the figure.

Revised -Figure 3

Reviewer 3 Report

Comments and Suggestions for Authors

The authors analyzed the metabolic profile of CSF in MSA. They constructed a network of altered CSF metabolites in MSA and examined these changes using bioinformatics approaches. Their findings indicate elevated ROS, cytoplasmic inclusions, white matter demyelination, ataxia, and neurodegeneration, accompanied by suppressed ATP concentrations, neurotransmitter release, and oligodendrocyte counts in MSA CSF samples. The authors concluded that integrated multiomics and bioinformatics approaches are crucial for elucidating neurodegenerative mechanisms and identifying potential diagnostic biomarkers for MSA.

While some aspects of their comprehensive analyses are intriguing, the study presents certain issues.

1.       While the authors noted the challenge of differentiating MSA from other synucleinopathies, they did not attempt to do so using the metabolic profile of the CSF.

2.       The primary limitation of this study lies in the recruitment process. While it appears that patients diagnosed with MSA were included, it is unclear which clinical criteria were applied or whether the diagnoses were pathologically confirmed.

3.       Another limitation lies in the process of collecting CSF samples. Since metabolite levels are highly influenced by the procedures used during sample collection, it would be important to understand how the authors ensured the quality and consistency of the samples.

4.     MSA is traditionally divided into two subtypes: MSA-C and MSA-P. How do these clinical differences influence CSF metabolite profiles?

Author Response

Answers to Reviewer 3

Thank you for your extremely insightful and valuable feedback. We have carefully considered and addressed each aspect, revising the manuscript accordingly. Below, we present a structured response, with your comments in black, our replies in blue, and the changes made to the manuscript highlighted in red.

Major Revision

Comment 1 - While the authors noted the challenge of differentiating MSA from other synucleinopathies, they did not attempt to do so using the metabolic profile of the CSF.

Response 1 – This study represents the first attempt to predict MSA using both metabolomic and proteomic datasets. Given the limited availability of metabolites, distinguishing MSA from other synucleinopathies based on metabolomics alone is challenging. Few studies compare MSA to other synucleinopathies rather than just healthy controls, and many known metabolites have been extensively studied. Future research should explore novel metabolites and employ larger datasets for more robust in silico predictions.

Revised – Lines 50 -55

Elevated  levels of the free fatty acid eicosapentaenoic acid (EPA) were observed in the CSF of both MSA and PD patients compared to healthy controls. However, no significant difference was found between MSA and PD, emphasizing the need for further metabolomic studies to distinguish MSA from other synucleinopathies which is critical for implementing appropriate treatment strategies at the onset of each disease.

Comment 2 - The primary limitation of this study lies in the recruitment process. While it appears that patients diagnosed with MSA were included, it is unclear which clinical criteria were applied or whether the diagnoses were pathologically confirmed.

Response 2 – We acknowledge the importance of clearly outlining the diagnostic criteria for the recruited MSA patients  and have revised our manuscript to accommodate your feedback.

Revised –Lines 418 -421

Firstly, since the most definitive confirmation of MSA is obtained through autopsies, patients were diagnosed based on clinical criteria widely recognized as defining features of MSA. Therefore, as this study primarily utilized CSF samples, pathological confirmation of MSA was not possible.

Comment 3 - Another limitation lies in the process of collecting CSF samples. Since metabolite levels are highly influenced by the procedures used during sample collection, it would be important to understand how the authors ensured the quality and consistency of the samples.

Response 3 – We agree with your comment and have made the following changes to our manuscript incorporating the limitation in maintaining quality and consistency of samples.

Revised – Lines 424 - 425

Additionally, the quality and consistency of samples were not standardized independently as they were obtained from contemporary reports.

Comment 4 - MSA is traditionally divided into two subtypes: MSA-C and MSA-P. How do these clinical differences influence CSF metabolite profiles?

Response 4 – Abdo et al (2007) (https://pubmed.ncbi.nlm.nih.gov/17448720/) studied CSF biomarker profiles to investigate whether the variation in clinical expression between MSA-P and MSA – C impacted the neurochemical profile and reported no distinction between the two sub types of MSA based on neurochemical biomarkers. Several other studies that we have referred to haven’t distinguished any variation in neurochemical profile between the two subtypes. However, we recognize the importance of your comment regarding influence of metabolic profiles on the two sub types.  Therefore, we have added the following revision to our manuscript.

Revised – Lines 106 - 114

MSA can be divided into two subtypes: multiple system atrophy Parkinson type (MSA-P) and multiple system atrophy cerebellar type (MSA-C). Primary symptoms of MSA-P resemble Parkinson’s disease such as tremors, rigidity and bradykinesia while MSA-C primarily affects the cerebellum impairing balance and coordination issues in patients. While most reports present MSA as a general disease and do not differentiate between the two sub types, Abdo et al attempted to differentiate between the two based on neurochemical profile and reported no distinction between MSA-P and MSA-C. This highlights the importance of additional studies required for a deeper understanding of MSA and distinguish between MSA-P and MSA-C.

Round 2

Reviewer 1 Report

Comments and Suggestions for Authors

The authors responded to all the issues I raised. I have no further concerns.

Reviewer 2 Report

Comments and Suggestions for Authors

The authors have addressed all major and minor concerns.

Reviewer 3 Report

Comments and Suggestions for Authors

I have completed my review, and the authors have adequately addressed all of my comments.